# The role of cytochrome *bc*₁ inhibitors in future tuberculosis treatment regimens

Clara Aguilar-Pérez [1] ✉, Anne J. Lenaerts[2], Cristina Villellas[1], Jerome Guillemont[3], John Dallow[4], Hannah Painter[4], Nicole C. Ammerman[5,6], Anis Hassan[4], Guillaume Golovkine [7], Laure Brock[7], Sylvie Sordello[7], Aurélie Chauffour [8], Alexandra Aubry[9], Thi Cuc Mai[8], Sarah Wong[8], Taane G. Clark [4], Kiyean Nam[10], Jeongjun Kim[10], Jinho Choi[10], Marjolein Crabbe[11], Jorge Esquivias[12], Nacer Lounis[1], Bart Stoops[11], Katie Amssoms[11], Jose M. Bartolome-Nebreda[12], Veronica Gruppo[2], Gregory T. Robertson [2], Nicolas Veziris[9], Anna M. Upton[13], Eric L. Nuermberger [5], Vivian Cox[14], Lluis Ballell[12], Benny Baeten[1], Anil Koul [1,4], Alexander S. Pym[15], Richard J. Wall [4,17] & Dirk A. Lamprecht [1,16,17] ✉

Tuberculosis (TB) remains the foremost cause of death from infectious diseases globally, prompting ongoing efforts to improve treatment options. This includes developing compounds with novel modes of action and identifying optimal treatment regimens that allow for treatment shortening. One promising strategy involves targeting cytochrome *bc*₁ oxidase in *Mycobacterium tuberculosis*, a key enzyme in the respiratory chain. In this study, we evaluate the potential of cytochrome *bc*₁ inhibitors as partner drugs in TB combination regimens. Using a relapsing mouse model, we demonstrate that these inhibitors enhance regimen sterilisation and significantly reduce the time required for effective treatment. We also propose several novel combination strategies for both multidrug-resistant and drug-sensitive TB, where cytochrome *bc*₁ inhibitors contribute to sterilisation and improved treatment outcomes. Furthermore, *M. tuberculosis* clinical isolates exhibited heightened susceptibility to cytochrome *bc*₁ inhibitors compared to laboratory-adapted strains, highlighting the importance of using clinical isolates in TB drug discovery to better reflect the diversity of TB populations. These findings emphasise the potential of cytochrome *bc*₁ inhibition in the development of more effective and shorter treatment regimens for TB, supporting the need for further clinical investigation.

Tuberculosis (TB), an infection caused by *Mycobacterium tuberculosis*, remains one of the deadliest infectious diseases, responsible for 1.25 million deaths in 2023[1]. The severity of the global TB pandemic is further exacerbated by the emergence of multidrug-resistant TB (MDR-TB). Effective treatment of MDR-TB typically requires a combination of three to five drugs, many of which are associated with toxicity and severe adverse events. These side effects often lead to poor adherence to treatment. Combination treatment is essential to shorten treatment duration, minimise the risk of drug resistance and prevent relapse. Nevertheless, current regimens still require many months of treatment with prolonged therapy contributing to reduced patient compliance and an increased risk of drug resistance.

Considerable research efforts are focused on optimising drug combinations that enable treatment shortening. For example, the TB-PRACTECAL clinical trial aimed to identify combination regimens, given as a 6 month oral regimen, capable of shortening treatment for DR-TB[2,3]. The final results demonstrated that the BPaLM (bedaquiline, pretomanid, linezolid and moxifloxacin) regimen was both safer and more effective, achieving more favourable outcomes (88%) in patients compared with the standard of care (SoC; 59%). Similarly, BPaL (86%) and BPaLC (bedaquiline, pretomanid, linezolid, clofazimine; 77%) regimens also resulted in higher proportions of MDR-TB patients with favourable outcomes, supporting the potential of shorter, optimised therapies using these compounds.

The World Health Organisation's consolidated 2024 guidelines now include new recommendations for MDR-TB treatment shortening[4,5]. These recommendations suggest two 6 month regimen options (compared with the previous 9 to 18 month duration): BPaLM for ages 14 years and older, and BDLLfxC (bedaquiline, delamanid, linezolid, levofloxacin and clofazimine) for children, adolescents, adults and pregnant women. However, the central role that fluoroquinolones and linezolid play in those regimens poses serious limitations due to drug resistance and toxicity, respectively. Moxifloxacin or levofloxacin must be excluded in cases of fluoroquinolone resistance, which is rapidly becoming a major challenge for treating MDR-TB[6]. This results in the modification of these regimens to BPaL and BDLC, respectively. Furthermore, the use of linezolid in MDR-TB treatment is often associated with adverse events, such as myelosuppression and peripheral neuropathy, which may negatively impact patient adherence. Thus, although MDR-TB treatment options are diversifying and improving[7], there is growing interest in identifying alternatives to the fluoroquinolones and linezolid to partner with BPa[8–11]. Furthermore, emerging drug resistance adds additional concerns, highlighting the urgent need to discover compounds with novel modes of action (MoA) and develop innovative therapeutic approaches.

The discovery of telacebec (Q203) and its successful Phase 2 A Early Bactericidal Activity (EBA) trial has established inhibition of the *M. tuberculosis* cytochrome $bc_1$ complex as a promising new drug target[12–15]. As a crucial component of the electron transport chain, essential for ATP production, cytochrome $bc_1$ represents an attractive partner target for other inhibitors of oxidative phosphorylation, such as bedaquiline. However, cytochrome $bd$, a terminal oxidase predominantly active in low-oxygen environments, has been identified as a critical bypass mechanism that can sustain bacterial respiration under cytochrome $bc_1$ inhibition[16]. This has led to growing interest in dual inhibition strategies and positioned cytochrome $bd$ as another underexploited target in TB drug development. While cytochrome $bc_1$ inhibitors have demonstrated considerable potential, their specific role within future TB treatment regimens is not clearly defined[17].

While counting colony-forming units (CFU) can provide a measure of bacterial burden during treatment, it does not capture the ability of a regimen to fully sterilise infection and prevent relapse. A relapse study, which assesses bacterial regrowth after treatment cessation, provides a more stringent and clinically relevant measure of treatment efficacy. This is particularly important in TB, where surviving *M. tuberculosis* bacilli may enter a non-replicating or persistent state and later cause disease recurrence. Therefore, relapse models offer critical insights into the sterilising activity of drug combinations, which is a key consideration for regimen design aimed at treatment shortening.

Here, using relapsing TB mouse models, we assessed the contribution of validated cytochrome $bc_1$ inhibitors, including the tool compound JNJ-2901[18], in different treatment regimen strategies to assess their potential to replace and mitigate the liabilities of the fluoroquinolones and linezolid. Next, we demonstrate that a cytochrome $bc_1$ inhibitor could contribute to an ultrashort treatment for drug sensitive TB (DS-TB), potentially contributing to a ≤2 month treatment regimen. Finally, we demonstrate that cytochrome $bc_1$ inhibitors are significantly more bactericidal against *M. tuberculosis* clinical isolates compared to laboratory-adapted strains, suggesting future drug discovery efforts should consider more emphasis on the testing of clinical isolates during drug development. Our findings suggest that cytochrome $bc_1$ inhibitors could provide an important contribution to TB treatment regimens by enhancing sterilisation, reducing relapse rates, and improving treatment outcomes.

## Results

### Cytochrome $bc_1$ inhibitors as alternative partner drugs in MDR-TB treatment regimens

Drug resistance to fluoroquinolones, along with the adverse events associated with linezolid, has prompted the search for alternative partner drugs for MDR-TB treatment. In this context, we recently identified JNJ-2901, an analogue of telacebec, as a promising candidate for further development of best-in-class inhibitors targeting cytochrome $bc_1$ (Supplementary Table S1–3). JNJ-2901 exhibited sub-nanomolar activity against MDR-TB strains and achieved a 4-log reduction in bacterial burden using a *M. tuberculosis* cytochrome *bd* knockout (CytBd-KO) strain in an acute mouse model of TB infection[18]. Cryo-electron microscopy structural analysis revealed that JNJ-2901 occupies the same binding pocket as telacebec and other structurally related cytochrome $bc_1$ inhibitors, confirming a shared MoA. All inhibitors described in this study were evaluated for their absorption, distribution, metabolism and excretion (ADME) properties, pharmacokinetics (PK) and toxicology profiles, supporting their further use in subsequent experiments[14] (Supplementary Table S2–3).

Using a relapsing mouse model, we compared the efficacy of cytochrome $bc_1$ inhibitors, including JNJ-2901 and telacebec, in various combination regimens (Fig. 1a, Supplementary Fig. S1). We first focused on replacing linezolid with JNJ-2901 (J) in the presence and absence of either moxifloxacin (M) or clofazimine (C) in the BPaL regimen, the recommended SoC for fluoroquinolone-resistant MDR-TB. Only BPaMZ provided a completely sterile cure after 8 weeks treatment (defined as 0/15 mice; 0%), measured after 12 weeks of relapse post treatment. However, BPaCJ (33%) was superior to BPaL (87%) based on relapse rates. When combined with JNJ-2901, the addition of clofazimine (BPaCJ; 33%) resulted in a statistically significant reduction in relapse rates compared to the addition of moxifloxacin (BPaMJ; 100%; $p = 0.003$) (Fig. 1b; Supplementary Table S4–5; Study A).

In a second relapse study, we focused on the impact of clofazimine in more detail (Fig. 1c; Supplementary Table S4, S6–7; Study B). BPaL led to a greater decrease in CFU after 8 weeks compared to BPaJ, but with comparable relapse rates observed after 8 weeks (100% for BPaL and BPaJ) and 12 weeks of treatment (27% for both regimens; Fig. 1c; Supplementary Table S4, S6–7). Replacing linezolid with clofazimine resulted in a similar decrease in CFU after 8 weeks but led to fewer relapses post-treatment (100% for BPaL, 27% for BPaC). BPaCJ demonstrated the best bactericidal effect, achieving an additional 1 $\log_{10}$ decrease in CFU compared to BPaC after 8 weeks, translating to a 33% relapse rate. No relapses occurred after 12 weeks of treatment with either BPaC or BPaCJ, highlighting the potential for treatment shortening compared to the BPaL SoC (Fig. 1b, c; Supplementary Table S4–7). These findings demonstrate that combining a cytochrome $bc_1$ inhibitor with BPaC increases the sterilising activity compared to BPaC alone, suggesting that cytochrome $bc_1$ inhibitors could play an important role in future MDR-TB regimens.

In an additional study (Supplementary Table S8–10; Study C), JNJ-2901 reduced the relapse rate when combined with the BPaL regimen, despite an inability to rescue mice when administered as a mono-therapy in an intravenous model using the H37Rv reference strain (Supplementary Fig. S2). There was a trend towards improvement of

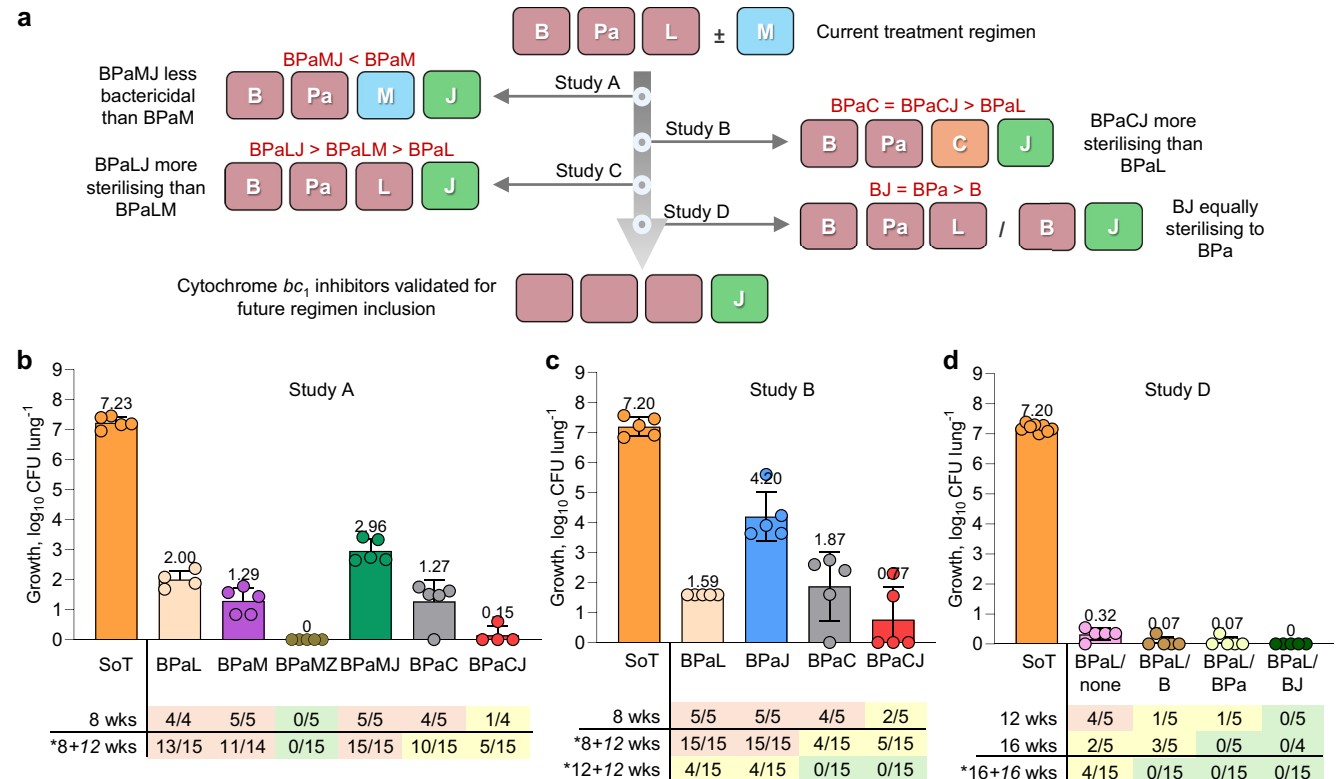

**Fig. 1 | Investigating alternative MDR-TB treatment regimens based on inclusion of a cytochrome $bc_1$ inhibitor in relapsing mouse models of TB.**
**a** Schematic of the studies presented in this project. **b–d** Lung bacterial burden, proportions of mice with positive cultures at end of treatment and relapse in *M. tuberculosis*-infected mice from Studies A, B and D. CFU data is shown for week 8 (**b**, **c**) and week 12 (**d**). Details of treatment doses can be found in Supplementary Tables S4 and S11. bedaquiline: B; pretomanid: Pa; clofazimine: C; linezolid: L; moxifoxacin: M; pyrazinamide: Z; JNJ-2901: J. SoT: Start of treatment. *: relapse rate; 12 or 16 weeks after treatment cessation ( + 12 or 16 wks). Relapse rate (n/N). Source data are provided as a Source Data file.

both the bacterial burden (BPaLJ: $0.95 \pm 1.1$ versus BPaL: $2.06 \pm 1.3$ $\log_{10}$ CFU lung$^{-1}$, $p = 0.04$) and a quantifiable improvement in relapse rates, that failed to reach statistical significance within the limitations of this trial design (50% versus 89%, $p > 0.05$). Notably, BPaLJ was comparable to BPaLM and BPaMZ (Supplementary Table S8–10). We recommend future studies include increased replicates to provide greater statistical power. Nevertheless, this suggests that BPaLJ could be considered as a potential treatment regimen irrespective of fluoroquinolone susceptibility.

### Sterilisation potential of cytochrome $bc_1$ inhibitors in an alternative dosing regimen

We further evaluated the role of cytochrome $bc_1$ inhibitors in an alternative dosing strategy, where mice were treated with BPaL for 8 weeks during an initial phase, followed by an additional 8 weeks of either BPa, BJ, B or no treatment in a continuation phase (Fig. 1d; Supplementary Table S11; Study D). The rationale for this study design was to reduce the time and overall amount of drug required to reach bacterial sterility. Bacterial burdens in lungs were assessed after 8, 12 and 16 weeks, and relapse rates were measured after 16 weeks of treatment plus 16 weeks following treatment cessation. Both the BPaL/ BPa and BPaL/BJ regimens trended towards lower relapse rates compared to BPaL without continuation treatment, although this was not statistically significant ($p = 0.09$). No colonies were detected after 12 weeks of treatment with BPaL/BJ, whereas with BPaL/B, colonies were detected from 3/5 mice at the end of treatment (Fig. 1d; Supplementary Table S11). These results suggest that inclusion of a cytochrome $bc_1$ inhibitor could enhance regimen effectiveness during the continuation phase of treatment with fewer drugs and a reduced overall drug burden.

### Inclusion of cytochrome $bc_1$ inhibitors can lead to treatment shortening

Next, we focused on an ultra-short treatment strategy for DS-TB based on drugs targeting the respiratory pathway. In the recent TRUNCATE-TB trial, a 2 month BZ-containing regimen was shown to be as effective as the current 6 month SoC (isoniazid, rifampicin, pyrazinamide, and ethambutol; HRZE)[19]. Furthermore, we previously demonstrated that an in vitro combination of bedaquiline, clofazimine and telacebec (T), which all target components of the electron transfer chain, resulted in rapid killing, suggesting a strong basis for an ultra-short treatment[20]. To further investigate this in vivo, we assessed the efficacy of regimens containing telacebec alongside other drugs targeting the respiratory pathway, compared to the current DS-TB SoC (HRZE; Study E). BCZ, CZT and BCZT regimens all demonstrated superior efficacy over HRZE after 8 weeks of treatment in reducing lung CFU burdens (Fig. 2, Supplementary Tables S12–13; Supplementary Data 1; Study E). Both BCZ and BCZT regimens achieved 0% relapse rates 12 weeks after 6- and 8 weeks of treatment, whereas CZT required at least 12 weeks of treatment to achieve 0% relapse. One of the mice treated with HRZE remained culture positive even after 20 weeks of treatment. The benefit of adding telacebec to BCZ is shown by the relapse rates after 4 weeks treatment: BCZ (60%) versus BCZT (36%), highlighting the treatment-shortening potential of cytochrome $bc_1$ inhibitors in combination with other drugs targeting the electron transport chain. Adding telacebec to the CZ core regimen reduced bacterial load by 2.2 $\log_{10}$ CFU and dramatically improved relapse rates after 12 weeks treatment from 100% (CZ) to 0% (CZT; $p < 0.0001$). Furthermore, CZT, BCZT and BCZ regimens significantly shortened treatment compared to HRZE (Fig. 2; Supplementary Tables S12–13; Supplementary Data 1). These findings provide strong evidence that incorporating a

cytochrome $bc_1$ inhibitor into new treatment regimens offers substantial benefits which should be further investigated.

## Increased susceptibility of cytochrome $bc_1$ inhibitors against clinical isolates

We found that the impact of cytochrome $bc_1$ inhibitors may be underrepresented depending on the *M. tuberculosis* strain selected for investigation. Here, using a range of cytochrome $bc_1$ inhibitors, including JNJ-4052, which has similar activity, ADME and PK profiles as JNJ-2901 (Supplementary Fig. S1; Supplementary Tables S1–3), we demonstrate that TB clinical isolates exhibited increased susceptibility to cytochrome $bc_1$ inhibition. This was observed against a diverse panel of clinical isolates from infected patients in both minimum inhibitory concentration (MIC) and minimum bactericidal concentration (MBC) assays, when compared to the laboratory-adapted H37Rv strain (Fig. 3a–d; Supplementary Table S14). This heightened susceptibility translated into increased in vivo efficacy in an acute mouse model, where a statistically significant CFU reduction (1.6 $\log_{10}$ reduction;

$p < 0.01$) was observed following cytochrome $bc_1$ inhibitor (JNJ-4052) treatment in mice inoculated with a clinical isolate (Fig. 3e; Supplementary Tables S1–S3; Study F). These results further emphasise the importance of including clinical isolates when evaluating the in vivo efficacy of new compounds.

## Discussion

Our findings highlight the potential of cytochrome $bc_1$ inhibitors as valuable components of future TB treatment regimens. These compounds may serve as effective alternatives to linezolid in SoC regimens for MDR-TB and fluoroquinolone-resistant MDR-TB. Given their novel MoA, cytochrome $bc_1$ inhibitors may also contribute to raising evolutionary barriers to resistance when used in combination and may be suitable for inclusion in DS-TB regimens to reduce treatment duration.

Substitution of linezolid with JNJ-2901 within BPaL regimens demonstrated promising results. While no regimen achieved complete sterilisation after 8 weeks of treatment, the BPaCJ combination significantly reduced relapse rates compared to BPaL and BPaMJ, demonstrating the additive benefit of clofazimine over moxifloxacin when paired with a cytochrome $bc1$ inhibitor. Replacing linezolid with clofazimine (BPaC) also led to improved post-treatment outcomes. The further addition of a cytochrome $bc_1$ inhibitor (BPaCJ) not only reduced bacterial burden more effectively than BPaC alone but also achieved relapse-free cure after 12 weeks, indicating enhanced sterilising activity and the potential for treatment shortening. Collectively, our findings support further evaluation of cytochrome $bc_1$ inhibitors as promising components of future MDR-TB regimens.

Effective TB treatment regimens require a combination of drugs with both bactericidal and sterilising activities to rapidly reduce bacterial loads and eliminate drug-tolerant bacilli. A two-phase regimen, with an initial intensive phase of drugs that rapidly reduce bacterial burdens, followed by sterilising drugs in a continuation phase, is a proven strategy for improving treatment outcomes for patients. Drugs in the continuation phase must target hard-to-treat drug-tolerant bacteria that survive the initial intensive phase[21]. In our study, the inclusion of a cytochrome $bc_1$ inhibitor in the continuation phase of the BPaL/BJ regimen demonstrated a sterilising effect similar to that of the BPaL/BPa regimen, suggesting that a cytochrome $bc_1$ inhibitor could serve as an effective alternative to pretomanid. Furthermore, a two-phase regimen could facilitate the development of long-acting injectable formulations, offering an alternative strategy to improve treatment adherence and outcomes[22].

There is an urgent need to develop ultra-short TB treatment regimens for drug-sensitive TB to improve patient compliance and reduce the risk of drug resistance. Bedaquiline-containing regimens have emerged as the most promising approach to achieve this goal. Additionally, a 14 day Phase IIA clinical trial previously demonstrated

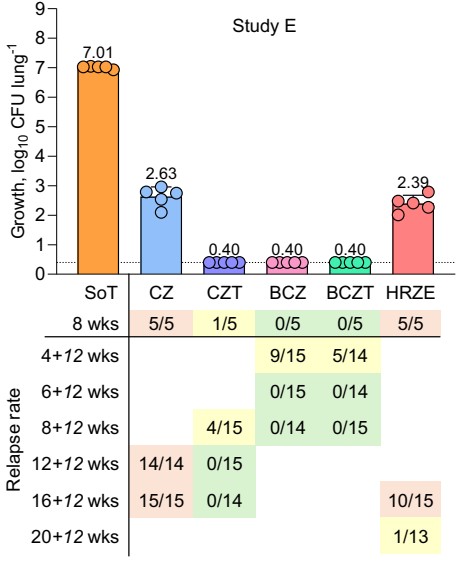

**Fig. 2 | Investigating alternative DS-TB treatment regimens based on inclusion of a cytochrome $bc_1$ inhibitor.** Lung bacterial burden and relapse in *M. tuberculosis*-infected mice from Study E. Details of treatment doses can be found in Supplementary Table S12. Limit of detection (dotted line) was 0.4 $\log_{10}$ CFU lung⁻¹. Bedaquiline: B; clofazimine: C; rifampicin: R; ethambutol: E; isoniazid: H; telacebec: T; pyrazinamide: Z. Source data are provided as a Source Data file.

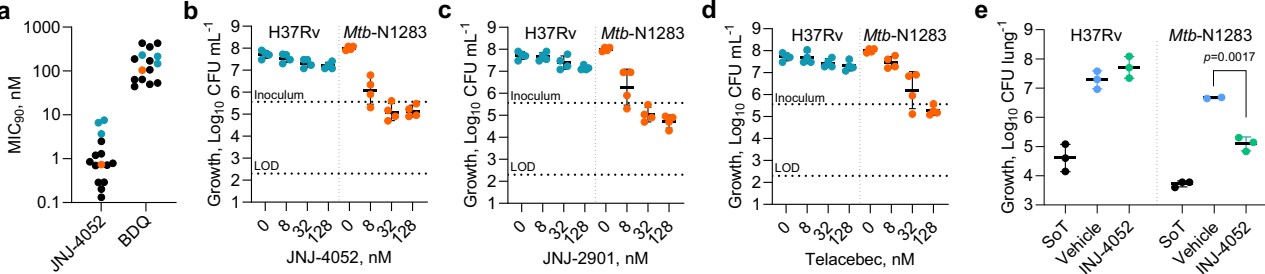

**Fig. 3 | Bactericidal activity of cytochrome $bc_1$ inhibitors in clinical isolates.** **a** Distribution of $MIC_{90}$ (concentration achieving 90% inhibition) values for a diverse selection of clinical isolates (black) compared with the lab-adapted WT H37Rv (cyan) for JNJ-4052 and bedaquiline (BDQ) (Supplementary Table S14). Clinical isolate N1283 (orange) is indicated. $n = 2$ technical replicates. Impact of (**b**) JNJ-4052, (**c**) JNJ-2901 and (**d**) telacebec on CFU counts in WT H37Rv compared to N1283 clinical isolate. $n = 4$ biological replicates for (**b**–**d**). **e** In vivo efficacy (Study F) of JNJ-4052 (50 mg kg⁻¹; PO) in H37Rv and N1283 after 2 weeks of treatment. SoT: Start of treatment, 7 days after inoculum with 300 CFU. $n = 3$ mice. Significance calculated with an ordinary one-way ANOVA. Data are presented as geometric mean ± geometric SD (panels **b**–**d**) or mean ± SD (**e**). Source data are provided as a Source Data file.

that pyrazinamide enhances the efficacy of bedaquiline-containing regimens, thereby contributing to treatment shortening[23]. Here, we show that the inclusion of a cytochrome $bc_1$ inhibitor dramatically reduced treatment duration, particularly in comparison to the current SoC (HRZE). Furthermore, a BCZ-background, especially when enhanced with telacebec, showed potential to achieve treatment durations of under 2 months. Further evaluation in larger preclinical and clinical studies is required to confirm long-term efficacy and relapse rates.

An interesting finding from our study was the increased bactericidal activity of cytochrome $bc_1$ inhibitors against clinical isolates and a cytochrome $bd$ knockout strain. Similar results have recently been reported, showing that the clinical isolate *M. tuberculosis* HN878 is more susceptible to cytochrome $bc_1$ inhibitors[24]. This study demonstrated that the addition of telacebec to BPaC in HN878 led to a 2-log reduction in lung bacterial burden compared to BPaC alone after 4 weeks of treatment. This likely reflects differences in the expression of cytochrome $bd$ after cytochrome $bc_1$ inhibition. Clinical isolates are reported to have a lower basal level of expression of cytochrome $bd$ than the lab-adapted strain H37Rv[25]. Under low $bd$ expression or in knockout conditions, the bacterium is forced to rely more heavily on cytochrome $bc_1$ for respiration, making it more vulnerable to inhibition[16]. This is evident from the $MBC_{99.9}$ data for H37Rv, where cytochrome $bc_1$ inhibitors are primarily bacteriostatic, whereas in the H37Rv_CytBd-KO strain, the same inhibitors are bactericidal (Supplementary Table S2). Taken together, these findings suggest the choice of strain can influence the apparent drug activity and support the routine inclusion of clinical isolates in early drug discovery and regimen design studies.

While our study provides important insights into the activity of novel combinations containing cytochrome $bc_1$ inhibitors, there are several considerations that can inform future experimental design. First, some experiments were underpowered to detect statistically significant differences between conditions. Future studies will increase animal numbers to strengthen statistical confidence and maximise interpretability. Second, we did not include monotherapy arms for each compound in the combinations tested, which limits our ability to deconvolute the contribution of individual agents. Including monotherapy controls in future work will provide a clearer understanding of synergistic or additive effects. Third, only female mice were used in this study to simplify animal husbandry and reduce stress-related variables as male mice require separation and can experience negative effects from isolation, increasing welfare concerns and experimental attrition. While sex-based differences have been reported in clinical outcomes[26], and some sex-specific immunological differences have been reported for different infectious diseases[27,28], gender bias remains unclear when studying the therapeutic effect of antitubercular drugs. As our goal was to provide proof-of-concept data rather than clinical validation, we chose a single-sex design for consistency, reproducibility and humane conduct of our multi-month studies; however, future studies should explicitly investigate potential sex differences, particularly when advancing regimens toward clinical development. Finally, our findings highlight that clinical isolates can exhibit markedly different susceptibility profiles compared to the widely used H37Rv lab-adapted strain. Future work should consider including diverse clinical strains to better capture the variability seen in patient-derived *M. tuberculosis* populations and improve translational relevance.Building on the established early bactericidal activity and novel mechanism of action of telacebec, cytochrome $bc_1$ inhibitors continue to show promise as key components in future TB treatment regimens. Their ability to enhance sterilising activity, particularly in combination with bedaquiline and clofazimine, supports their potential role in ultra-short TB treatment regimens. Beyond TB, telacebec is also under investigation for the treatment of Buruli ulcer and leprosy[29–31].

Cytochrome $bc_1$ inhibitors not only improved the bactericidal and sterilising activity of regimens in mouse relapse models but also showed enhanced potency against clinical isolates. Our data highlight the potential of both clofazimine- and cytochrome $bc_1$ inhibitor-containing regimens to significantly shorten treatment durations for both DS- and DR-TB. This work serves as a foundation for further exploration of other treatment regimens incorporating cytochrome $bc_1$ inhibitors and there are several in vitro, ex vivo and in vivo tools available to expedite this process. For example, pairwise drug interactions can be rapidly assessed using DiaMOND analysis to identify optimal combinations in vitro[32]. These combinations can then be tested in ex vivo caseum models to investigate the contribution of cytochrome $bc_1$ inhibition against non-replicating, persistent bacteria[33]. Promising regimens could subsequently be evaluated in an ultra-short-course treatment model that incorporates RS-ratio measurements to rapidly assess bactericidal activity and sterilising potential[34].

These findings support the inclusion of cytochrome $bc_1$ inhibitors in future regimen design efforts and highlight the need for further studies to define optimal dosing strategies, assess long-term relapse outcomes and evaluate their role in additional combinations.

## Method

Treatment regimens were evaluated in five studies (Studies A–E), in a relapsing mouse infection model of TB treatment[35]. Animal studies were performed at Evotec France SAS Toulouse (Studies A and B), Sorbonne University (Study C)[36], Johns Hopkins University (Study D) and Colorado State University (Study E), in accredited facilities. An additional study (Study F) in an acute mouse infection model was performed at London School of Hygiene & Tropical Medicine[37,38].

### Ethical statement

In Studies A and B, animal experiments were performed under the European Union Directive and with local ethical committee clearance. The study procedures were reviewed by the Evotec France Ethical Committee and authorised by the French Ministry of Education Advanced Studies and Research. For Study C, the experimental project was favourably evaluated by the ethics committee n°005 Charles Darwin localised at the Pitié-Salpêtrière Hospital and clearance was given by the French Ministry of Education and Research under the number APAFIS#12380-2017112809414820 v3. The animal facility received the authorisation to carry out animal experiments (license number C-75-13-08). For Study D, ethics oversight was provided by the Johns Hopkins University Animal Care and Use Committee, which is PHS assured, USDA registered, and AAALAC accredited. For Study E, ethics oversight was provided by the Colorado State University Animal Care and Use Programme (reference number, KP 5172) which is PHS assured, USDA registered, and AAALAC accredited. For Study F, animal procedures were performed under UK Home Office project license P6CA9EB8D and approved by the London School of Hygiene & Tropical Medicine Animal Welfare Ethical Review Board. All work was conducted in accordance with the UK Animal Scientific Procedure Act (ASPA) 1986. In all cases, the humane end points were: (i) body weight loss ($\geq 20\%$ weight loss), (ii) body condition scoring (behavioural changes), (iii) dehydration and (iv) laboured respiration (potentially due to exacerbated lung pathology).

### Animal housing

Female BALB/cJRj mice (studies A, B, D, E, and F) and female Swiss mice (Study C) were housed socially in bio-confined BSL3 cages (up to 5 animals/cage) under a 12-h light: 12-h dark with access to filtered water and a standard rodent diet ad libitum. An ambient temperature of $22 \pm 2\,°C$, a relative humidity of $55 \pm 10\%$ and a negative pressure of -20Pa were maintained. All mice were allowed to acclimatise to their new environment for at least 5 days after identification. Mice were observed daily.

## Study design

In Studies A and B, mice were infected by intranasal inoculation of 50 μL of *M. tuberculosis* H37Rv at inoculum level of 4.5 $\log_{10}$ CFU mouse$^{-1}$. Treatment was initiated 2 weeks post-infection, when the bacterial burden in the lungs was >7.19 $\log_{10}$ CFU. The mice were treated 5 days per week by oral gavage (at 10 mL kg$^{-1}$), for 4, 8, and 12 weeks and received different treatment combinations (Supplementary Table S4) containing bedaquiline (25 mg kg$^{-1}$), pretomanid (40 mg kg$^{-1}$), linezolid (100 mg kg$^{-1}$), clofazimine (20 mg kg$^{-1}$), and JNJ-2901 (5 mg kg$^{-1}$). For all groups, the first two drugs were administered in the morning and the last two drugs in the afternoon, with around 2 h between the administration of each drug. The bacterial load in lungs (CFU lung$^{-1}$) was assessed at the end of each treatment, after a standard washout period of 3 days, in 5 animals/group. An untreated control group consisted of 10 mice (5 at D-13 to act as infection control and 5 at D0 to determine the infection level at treatment start). At the end of 8 or 12 weeks of treatment, 15 animals per group were held without treatment for 12 weeks to determine the proportion of mice with relapse. The total number of mice was 160 in Study A and 255 in Study B. Other study endpoints included animal weights weekly prior to treatment and post treatment during the relapse period and 3 times weekly during treatment phase, lung weight 1 day and 14 days pi and at the end of each period of treatment and relapse period, and observation of treatment-emergent adverse effects.

For Study C (Supplementary Table S8), the *M. tuberculosis* reference strain H37Rv maintained for animal infection was grown on Löwenstein-Jensen medium for 3 weeks. Colonies were subcultured in 7H9 medium supplemented with 10% OADC for 7 days at 37 °C. 112 six-week-old female Swiss mice were purchased from Janvier breeding centre. Mice were infected intravenously with 0.5 mL of the bacterial suspension of 6.1 $\log_{10}$ CFUs. Mice were treated orally 5 days per week for 3 months with the following drugs (dosing in mg/kg/day): bedaquiline (B, 25 mg kg$^{-1}$), JNJ-2901 (J, 10 mg kg$^{-1}$), moxifloxacin (M, 100 mg kg$^{-1}$), pretomanid (Pa, 40 mg kg$^{-1}$), linezolid (L, 100 mg kg$^{-1}$), rifampicin (R, 10 mg kg$^{-1}$), isoniazid (H, 25 mg kg$^{-1}$) and pyrazinamide (Z, 150 mg kg$^{-1}$). Six untreated mice were sacrificed the day after infection (D-13), the day of treatment initiation (D0); 20 mice were allocated in each arm and were euthanized after 16 weeks of treatment and 12 weeks off treatment for relapse rate assessment.

In Study D, 4–6 weeks old mice were infected by high-dose aerosol with *M. tuberculosis* H37Rv (~4 $\log_{10}$ CFU mL$^{-1}$), using a Glas-Col Inhalation exposure system. Treatment started at 2 weeks pi (when the bacterial burden was >7 $\log_{10}$ CFU) and was administered by oral gavage, once daily, 5 days per week. Mice received different treatment combinations (two or three drugs; Supplementary Table S11) containing bedaquiline (25 mg kg$^{-1}$), pretomanid (100 mg kg$^{-1}$), linezolid (100 mg kg$^{-1}$) and JNJ-2901 (5 mg kg$^{-1}$). All treatment groups received BPaL for an initial 8-week intensive phase, then received treatment with B alone or in combination with either Pa or J, or no treatment, for an 8-week continuation phase. The untreated control group consisted of 26 mice (eight at D-13, eight at D0, and 5 at each 2- and 4-weeks after the start of treatment). Lung CFU was assessed after 8, 12 and 16 weeks of treatment (5 mice/group), as well as at the end of the relapse period (15 mice/group).

In Study E, 6–8 weeks old mice were infected by high-dose aerosol with *M. tuberculosis* Erdman, using a Glas-Col Inhalation exposure system. Treatment was initiated at 11 days pi and was administered 5 days per week, via oral gavage at 200 μL/mouse, ensuring at least 1 h between the administration of regimen components. Mice received different treatment combinations (2–4 drugs; Supplementary Table S12) containing bedaquiline (25 mg kg$^{-1}$), clofazimine (20 mg kg$^{-1}$), pyrazinamide (150 mg kg$^{-1}$), telacebec (10 mg kg$^{-1}$), isoniazid (10 mg kg$^{-1}$), rifampicin (10 mg kg$^{-1}$), and ethambutol (100 mg kg$^{-1}$) for 4, 6, 8, 12, 16, and 20 weeks. The untreated control group consisted of 11 mice (six sacrificed at D-11 and 5 at the start of treatment on D0). Lung CFU was assessed at the end of the treatment period in 5 mice/group after a washout period of 5 days. Relapse groups were kept without treatment for 12 weeks; 15 mice/group were allocated for each arm. Mice were euthanised at the sampling time-points indicated in Supplementary Table S12.

In Study F, 6–8-week-old BALB/c mice (Charles River UK) were infected by high-dose aerosol with 300 CFU of either *M. tuberculosis* H37Rv mouse passaged or a clinical isolate, N1283. After 7 days, treatment was initiated with once-daily JNJ-4052 (50 mg kg$^{-1}$; PO) for 2 weeks. Lung CFU was assessed at the start of treatment (day 7) and after treatment (day 21) (3 mice/group).

## Drug formulation

Bedaquiline (salt-base ratio 1.21; studies A and B: LTK Laboratories, B165121. Study C, D and E provided by Janssen Pharmaceutica) was prepared weekly. It was dissolved in 20% 2-hydroxypropyl-β-cyclodextrin (Aldrich 332593) at pH 3 (adjusted with 1 N HCL), vortexed for 10 min and stirred overnight at 4 °C, protected from light. Pretomanid (Chemshuttle 140130), made fresh every week, was dissolved in 10% 2-hydroxypropyl-β-cyclodextrin (Aldrich 332593) 2% soy lecithin. Vortexed for 10 min and stirred overnight at 4 °C. Moxifloxacin (LTK Laboratories M5794), made fresh every day, was dissolved in water by vortexing for 1 min. Stirred at room temperature until complete dissolution. Clofazimine (LTK Laboratories C458567), made fresh each week, was weighed and transferred to a mortar. The powder was ground with a small volume of 20% 2-hydroxypropyl-β-cyclodextrin (Aldrich 332593), then mixed with 20% 2-hydroxypropyl-β-cyclodextrin. Compound was kept at 4°C and protected from light. Linezolid (LTK Laboratories L3453), made freshly each week, was weighed and transfer to a mortar. The powder was ground with a small amount of PEG-200 (5% of the total liquid volume; Sigma P3015). Transferred to a conical tube and vortexed. 0.5% methylcellulose (95% of the total liquid volume; Sigma M0430) was added and placed in a tube with slow rotation overnight. Compound was kept at 4 °C and protected from light. Pyrazinamide (ACROS 157641000), made fresh each week, was dissolved in sterile ddH$_2$O. The solution was heated to 60 °C until crystals dissolved. Compound was then kept at 4 °C and protected from light. Rifampicin (Sigma R3501), made fresh each week, was ground in a small amount of sterile water. Water was added in small increments while grinding until final volume/concentration was achieved. Compound was kept at 4 °C and protected from light. Ethambutol (Sigma E4630-25G), made fresh weekly, was dissolved in water then kept at 4 °C. Isoniazid (Sigma I3377-50G), was made fresh each week by dissolving in water then kept at 4°C. JNJ-2901 and JNJ-4052 (provided by Janssen Pharmaceutica), were made fresh each week, by dissolving in PEG400 (Aldrich P3265). Compounds were stirred until full dissolution and kept at room temperature. Telacebec (provided by Janssen Pharmaceutica) was made fresh each week by dissolving in PEG400 (Aldrich P3265). 12 N HCl was added (0.36 μL 12 N HCl per mL solution of 2 mg mL$^{-1}$ telacebec). Compound was stirred overnight and kept at room temperature. In studies A, B, C and D, drugs were administered individually with at least 2 h between dosages. In Study E, individual drugs were made as 2-3x concentrated stocks and combined and dosed together in a 0.2 mL volume at the time of dosing. Specifically: clofazimine and pyrazinamide were made separately at 2 times final concentration at half volume and combined at the time of dosing. Bedaquiline and clofazimine made separately at 2 times final concentration at half volume and combined at the time of dosing. Bedaquiline, clofazimine and pyrazinamide were made separately at 3 times final concentration at one-third volume and combined at the time of dosing. Isoniazid, ethambutol and pyrazinamide were made separately at 3 times final concentration at one-third volume and combined at the time of dosing.

## Pharmacokinetics (PK) and tolerability in mouse

The PK of JNJ-2901 and JNJ-4052 was investigated in female Balb-c mice dosed as solution or a suspension at 5 mg kg$^{-1}$ PO. Three animals were used. Animals had free access to food and water through each study. Blood samples were taken at multiple timepoints up to 24 h after PO dosing. Plasma samples were prepared by protein precipitation with acetonitrile, and the supernatant was analysed for concentrations of compound using a qualified LC-MS/MS method. Individual plasma concentration-time profiles were subjected to a non-compartmental pharmacokinetic analysis (NCA) using Phoenix.

The PK of JNJ-2901 was investigated in female CD-1 mice dosed as solution or a suspension at 5 mg kg$^{-1}$ PO and JNJ-4052 investigated in male CD-1 mice dosed as solution at 5 mg kg$^{-1}$ PO. Three animals were used. Animals had free access to food and water through each study. Blood samples were taken at multiple timepoints up to 24 h after PO dosing. Plasma samples were prepared by protein precipitation with acetonitrile, and the supernatant was analysed for concentrations of compound using a qualified LC-MS/MS method. Individual plasma concentration-time profiles were subjected to a non-compartmental pharmacokinetic analysis (NCA) using Phoenix.

## ADME assays

Details of the metabolic stability in liver microsomes and hepatocytes together with CYP inhibition in human liver microsomes and plasma protein binding have been published previously[39]. The CYP induction assay was conducted by Puracyp Inc (www.puracyp.com). Studies were designed to evaluate the effect of compounds on the activation of human PXR (hPXR). The DPX2TM cells harbour the human PXR gene (NR1I2) and a luciferase reporter gene linked to two promoters identified in the human CYP3A4 gene, namely XREM and PXRE. These cells were seeded in a 96-well plate. Twenty-four hours after seeding, the cells were treated with various concentrations of compounds then incubated for an additional 24 h. Following this, the number of viable cells per well were determined using Promega's CellTiter Fluor Cytotoxicity Assay. Briefly, One-Glo was added to the same wells and reporter gene activity was assessed. The luminescence light intensity was directly proportional to the extent of PXR activation and accompanying gene transcription in the DPX2TM cells.

## Enumeration of CFU from lung

In Studies A, B and F samples in Gentle Macs tubes were homogenised in phosphate-buffered saline (PBS) + 10% bovine serum albumin (BSA). Homogenates were plated pure 500 μL and 50 μL or serial diluted (50 μL) by 1/5 or 1/10 in PBS-BSA 10%. Dilutions were plated on 7H11-OADC plates containing 0.4% activated charcoal and incubated at 37 °C for CFU quantification using an automatic SCAN-1200 counter (Interscience). For relapse assessments, the total lung homogenate was plated undiluted. Reads were performed after 4 and 6 weeks of incubation at 37 °C. For each sample, remaining volumes were stored in case additional plating should be required. In Study F, the reads of the CFU were done manually.

In Study C, samples (whole lungs) were homogenised by using a GentleMacs Octo Dissociator (Miltenyi) under a volume of 2 mL physiological serum. The suspension was entirely plated on 7H11-OADC plates containing 0.4% activated charcoal. Reads were performed after 4 and 6 weeks of incubation at 37 °C.

In Study D, samples (whole lungs) were homogenised in 2.5 mL PBS, diluted in PBS in 1:10 steps and plated on 7H11-OADC plates containing 0.4% activated charcoal. After 8 weeks of treatment, 0.5 mL of each undiluted homogenate was plated in duplicate to capture low CFU counts. For all subsequent time points, at least 80% of the total homogenate volume was plated undiluted on 7H11-OADC agar plates with activated charcoal. Reads were performed after 4 and 6 weeks of incubation.

In Study E, samples (whole lungs) were homogenised in 4.0 mL PBS-BSA; half of each sample was stored in 0.05% tween-80 and 20% glycerol (f.c.). Homogenates were diluted in PBS-BSA in 1:5 steps and plated on 7H11-OADC plates containing 0.4% activated charcoal. 0.5 mL of each homogenate was also plated in duplicate to capture low CFU counts. For relapse assessments, tissues were homogenised in 1X PBS and one-half of the total volume was plated in its entirety on 7H11-OADC agar plates without activated charcoal. Remaining volumes were stored as above, in case additional plating was required. CFUs were enumerated after ≥6 weeks of incubation in sealed ziptop bags at 37 °C.

## Clinical isolates collection

The original Karonga study and follow-up work was approved by the London School of Hygiene & Tropical Medicine ethics committee (#5067) and the Health Sciences Research Committee in Malawi (#424). Consent was obtained for three sputum samples collected from each patient. *M. tuberculosis* isolates were cultured in liquid culture (in the absence of antimicrobial drugs) from frozen stocks of Lowenstein-Jensen or liquid cultures obtained from patient sputum isolates[40]. The Gagneux strains (N1283, N0072 & N0054) were purchased from the Belgian Coordinated Collections of Microorganisms/ Institute of Tropical Medicine (BCCM/ITM).

## Determination of minimum inhibitory/bactericidal concentrations (MIC/MBC)

Compounds were tested in 2-fold dilutions against *M. tuberculosis* strains ($5 \times 10^5$ CFU mL$^{-1}$) diluted in Middlebrook 7H9 medium supplemented with 10% OADC, 0.2% glycerol and 0.05% Tween 80. Plates for MIC determination were incubated for 7 days at 37 °C. Absorbance (OD$_{600}$) was measured using an FLUOstar Omega3 (BMG Labtech). MIC$_{90}$ values were determined as the drug concentration achieving 90% inhibition when compared with the antibiotic-free control wells. MBCs were determined by preparing a 4-fold serial dilution of each drug at x100 concentration in DMSO, before dilution with the bacterial inoculum. Plates were incubated for 18 days before serial dilution was prepared and 5 μL aliquots were plated onto 7H10 plates (supplemented with 0.4% charcoal, 0.5% glycerol and 10% OADC) and incubated for $21 \pm 5$ days at 37 °C before counting CFU. The MBC$_{99.9}$ was defined as the concentration giving at least 3-log$_{10}$ reduction compared to the initial CFU.

## Mutant isolation and Whole genome sequencing (WGS)

To isolate resistant colonies, agar plates containing 100 x MIC$_{90}$ of JNJ-2901 and JNJ-4052 were inoculated with WT *M. tuberculosis* Δ*cydAB* ($1-5 \times 10^8$ CFU) to select resistant colonies. Individual colonies were re-plated in the presence of compound for ~3 weeks to confirm resistance. Genomic DNA from selected clonal survivors was extracted using Quick-DNA Fungal/Bacterial Miniprep Kit (Zymo Research). WGS libraries were prepared using the Nextera XT DNA Library Preparation Kit (Illumina). Sequencing was performed on an Illumina NextSeq 550 platform, generating paired-end 75 bp reads and targeting 5 M sequences per sample. The trimmed reads were mapped to the reference genome of *M. tuberculosis* H37Rv (GenBank accession number NC_000962.3), and variants were identified using the CLC Genomics Workbench v21.0.5 (Qiagen) variant caller, with a minimum count of 2, minimum coverage of 10 and a minimum frequency of 10%. Variants present in all samples and wild type sample were filtered out. Only variants present in >75% of the reads were considered to minimize noise.

## Statistical analysis

Mean CFU lung$^{-1}$ was log-transformed before analysis and evaluated by one-way analysis of variance (ANOVA), followed by a pairwise multiple comparison using Dunnett's test or Tukey's test. The Kruskal-Wallis

one-way ANOVA on ranks or equal variance tests were used if data failed normality. The proportions of mice relapsing were compared using Fisher's exact test and interpreted with and without Bonferroni correction. Differences were considered significant at the 5% significance level. All analyses were carried out using GraphPad Prism software program.

## Reporting summary

Further information on research design is available in the Nature Portfolio Reporting Summary linked to this article.

## Data availability

All data generated or analysed during this study are included in this published article (and its supplementary information files). Source data are provided with this paper.

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

## Acknowledgements

The authors would like to thank: Amit Kaushik for technical support at Johns Hopkins University; AT Henze and PM Petrar for medical writing support at Janssen Pharmaceutica; Annelies Wouters and Lieve Lammens for toxicology input at Janssen Pharmaceutica; Nadia Neto for her feedback on WGS at Janssen Pharmaceutica; Gregory Bancroft for his intellectual input at the London School of Hygiene & Tropical Medicine; and Courtney I. Hastings at Colorado State University for technical support. This work was supported by Janssen Pharmaceutica NV, including all costs associated with the development and publishing of the manuscript. The work at the London School of Hygiene & Tropical Medicine was supported by funding from Janssen Pharmaceutica (AK and RJW). RJW is also funded by the UKRI via the Liverpool School of Tropical Medicine (MRC IAA21127). This project has received funding from the Innovative Medicines Initiative 2 Joint Undertaking under grant agreement No 853903 (RespiriTB). This Joint Undertaking receives support from the European Union's Horizon 2020 research and innovation programme and EFPIA. TGC is funded by the UKRI (BBSRC BB/X018156/1; MRC MR/X005895/1; EPSRC EP/Y018842/1).

## Author contributions

C.A.-P., D.A.L., A.J.L., S.S., G.T.R., N.L., A.U., N.V., E.L.N., N.C.A., A.A., A.C., B.B., B.S., H.P., J.D. and A.S.P. were involved in the conception or the design of the study. J.G. and JMB-N designed and coordinated the synthesised of the compounds. JE was involved in the chemical characterisation. A.K., C.V., D.A.L., B.B., A.S.P. and LBallell supervised the overall research programme. C.A.-P., S.S., E.L.N., A.C., N.C.A., T.C.M., S.W., N.V., N.W., H.P., J.D., A.H., B.S., LBrock, G.G. and G.R. participated in the collection or generation of the studies data. S.S., E.N., A.C., T.C.M., S.W., H.P., J.D., A.H., V.G. and G.T.R. performed the studies. C.A.-P., S.S., E.L.N., J.E., K.N., J.K., J.C., T.G.C. and G.R. contributed to the study with materials/analysis tools. C.A.-P., A.J.L., N.V., S.W., G.T.R., S.S., M.C., E.L.N., N.C.A., LBallell, J.D., H.P., V.C., B.S., R.J.W. and D.A.L. were involved in the analysis or interpretation of the data. C.A.-P., R.J.W. and D.A.L. wrote the manuscript. All authors reviewed drafts of the manuscript and gave final approval to submit for publication. All authors attest they meet the ICMJE criteria for authorship.

## Competing interests

J.G., C.V. and D.A.L. have been named inventors in a patent application for JNJ-2901 and JNJ-4052 compounds. C.A.-P., C.V., J.G., M.C., J.E., N.L., B.S., J.M.B.-N., V.C., LBallell, B.B., A.K., A.S.P. and D.A.L. were/are all full-time employees of Janssen, a Johnson & Johnson company, and/or potential stockholders of Johnson & Johnson. G.G., LBrock, S.S. and A.U. were/are all full-time employees of Evotec. K.N., J.K. and J.C. were/are full-time employees of Qurient Co. Ltd. J.D., H.P., A.H., A.K., A.J.L., G.T.R., N.C.A., V.G., E.L.N. and R.J.W. received funding from Janssen Pharmaceutica to perform contract research. The other authors declare no competing interests.

## Additional information

[1]Janssen Global Public Health, LLC, Janssen Pharmaceutica NV, Turnhoutseweg 30, Beerse, 2340 Antwerpen, Belgium. [2]Mycobacteria Research Laboratories, Department of Microbiology, Immunology and Pathology, Colorado State University, Fort Collins, Colorado 80521, USA. [3]Janssen Infectious Diseases Discovery, Janssen-Cilag, Val de Reuil, France. [4]Department of Infection Biology, Faculty of Infectious and Tropical Diseases, London School of Hygiene and Tropical Medicine, LondonWC1E 7HTUK. [5]Center for Tuberculosis Research, Department of Medicine, Johns Hopkins University, Baltimore, MD, USA. [6]Erasmus MC, University Medical Center Rotterdam, Department of Medical Microbiology and Infectious Diseases, Rotterdam, Netherlands. [7]Translational Biology, Infection Diseases, Evotec, 195, Route D'Espagne, 31100 Toulouse, France. [8]Sorbonne Université, INSERM, Centre d'Immunologie et des Maladies Infectieuses, U1135 Paris, France. [9]Sorbonne Université, INSERM, Centre d'Immunologie et des Maladies Infectieuses, U1135, APHP Sorbonne Université, Centre National de Référence des Mycobactéries et de la Résistance des Mycobactéries aux Antituberculeux, Paris, France. [10]Qurient Co. Ltd. C-dong 801, 242, Pangyo-ro, Bundang-gu Seongnam-si, Republic of South Korea. [11]Janssen Research & Development, LLC, Janssen Pharmaceutica NV, Turnhoutseweg 30, 2340 Beerse, Antwerpen, Belgium. [12]Discovery Chemistry, Janssen-Cilag SA a Johnson & Johnson company, C. Río Jarama, 75A, 45007 Toledo, Spain. [13]Evotec US inc., 303B College Road East, Princeton, NJ 08540, USA. [14]Johnson & Johnson Innovative Medicine, Titusville, NJ, USA. [15]Janssen Global Public Health, LLC, Janssen Pharmaceutica, 50-100 Holmers Farm Way, High Wycombe HP12 4DP, UK. [16]Present address: Holistic Drug Discovery and Development (H3D) Centre, University of Cape Town, Rondebosch 7700, South Africa. [17]These authors contributed equally: Richard J. Wall, Dirk A. Lamprecht. ✉e-mail: CAguila2@its.jnj.com; dirk.lamprecht@uct.ac.za

