## [Transparent Peer Review file · Nature Communications]

The role of cytochrome bc1 inhibitors in future tuberculosis treatment regimens

Corresponding Author: Dr Clara Aguilar Pérez

Version 0:

Reviewer comments:

Reviewer #1

(Remarks to the Author)

The manuscript by Perez and colleagues present data on the effects of cytochrome bc1 inhibitors and their potential efficacy in drug regimens against multidrug resistant TB as well as its role in reducing the duration of chemotherapy in drug susceptible TB. The work is important and timely considering the burden of drug-resistant TB and the prolonged treatment protocols for both drug resistant and susceptible cases of TB. The experimental approach is solid and data presented are robust to support their hypotheses.

One of the main interesting points in the manuscript is the fact that cytochrome bc1 inhibitors are more potent against clinical isolates than the lab strains. A further investigation into this would have increased the impact of the work as well as our understanding of the mechanisms of action of these inhibitors. Especially as these are claimed to be validated inhibitors, it would also be nice to see more of their characterisation and target engagement in vitro.

In addition, would this variability on strain efficacy mean that there are potential other targets being inhibited by the same compounds or is it the secondary effects of inhibiting the cytochrome bc1 and the respiratory chain of Mtb?

For the present study, authors can expand the discussion regarding the efficacy of their compound in the background of bd and how this could affect the outcome. More importantly, the authors can use these compounds to further interrogate this hypothesis. In addition, having another look at their data, I could see that if you calculate the MBC/MIC ratio for the reference lab strain (H37Rv) and the cytochrome deleted mutant (H37Rv_CytBd-KO) on supplementary table S2 one can see that their inhibitor is bacteriostatic on H37Rv whereas it is bactericidal on cytochrome bd deleted mutant (H37Rv_CytBd-KO). That further adds to the argument that the expression levels of cytochrome bd is an important factor in their compound efficacy.

Furthermore, the authors did not include if these compounds are active against MDR and XDR Mtb strains. Taking into account that the main use of these inhibitors would be towards a drug regimen for treating MDR-TB, it would be nice to see that the compounds maintain their activity against a panel of multiple drug-resistant strains including MDR and XDR clinical isolates.

Obviously a series of experiments where one looks at the levels of expression of these genes under treatment or not in a series of lab and clinical strains of Mtb would shed light on this question. But I don't think these experiments need to be part of this work.

Lastly, in the Figure S3, I cannot see the control line as well as the figure legend need a bit more explanation.

Reviewer #2

(Remarks to the Author)

The authors have done an excellent job of identifying cytochrome bc1 as an important component of the Mtb electron transport chain and a potentially sterilizing target in for TB regimen development. Using a relapsing mouse model, they have attempted to demonstrate that cytochrome bc1 inhibitors could be effective partner drugs in TB regimen development that could enhance sterilization. They report on several novel regimens

as examples of this role for bc1 inhibitors in regimens for both multidrug-resistant TB (MDR-TB) and drug-sensitive TB (DS-TB), where cytochrome bc1 inhibitors could contribute to sterilization and treatment shortening. Importantly, they have shown that clinical isolates exhibit heightened susceptibility to cytochrome bc1 inhibitors compared to laboratory-adapted strains, further supporting their potential usefulness in a clinical setting even as their contribution may be low or undetectable against a lab adapted strain. They take these findings as confirmation that cytochrome bc1 inhibitors have significant potential to improve TB treatment outcomes and highlight the need for further studies to evaluate their clinical contribution to novel treatment regimens.

What are the noteworthy results? The authors have done an excellent job of identifying and advancing highly potent inhibitors against their chosen target, cytochrome bc1. There is enough evidence to demonstrate that this is a sound choice of target, especially with the excellent success of bedaquiline, the flagship ATP synthase developed by the same group that has literally changed the paradigm in TB treatment shortening and highlighted the electron transport chain as a highly valuable and sterilizing target, at the same level or better than the rifamycins of prior years. It is therefore sound and admirable to hit this pathway as hard as possible, at multiple points and potentially achieve better cure. The problem seems to be that the interplay between cytochrome bc and cytochrome bd seems complicated and makes it difficult to determine with available data, including the data reported here, if just inhibiting bc, but not bd will be enough to realize the expected benefits. It seems that high expression of bd is able to reverse or nullify the effects of inhibiting bc. This is potentially the reason why there is strain difference between lab adapted versus clinical isolates. Further proof is gleaned from the excellent performance of Q203, the lead asset in this target space, versus mycobacterial species that do not express bd like *M. leprae* and *M. ulcerans*. This leaves the potential for a bc inhibitor uncertain against tuberculosis. Having said that, the team has identified a tool compound that together with Q203 will be useful in further investigating this target

Will the work be of significance to the field and related fields? While the actual developability of this asset in future TB drug regimens is in doubt until more work is done to fully appreciate contribution against all TB strains, publication of this work will be of great significance to the field and specifically to the grand idea of targeting the electron transport chain of *Mtb* at multiple nodes to achieve better cure for TB. Furthermore, the compound reported here could also be expanded to the NTM space as is being done for Q203 which is currently being evaluated against leprosy and Buruli ulcer, arguably important and neglected diseases as well. But for use in tuberculosis, more work needs to be done to understand the importance of targeting bc1 in a background with high expression of bd. And to determine if the site of infection and location in a lesion affects the usefulness of this asset. And if the mouse model as used today is the best model for this particular asset

How does it compare to the established literature? The work reported here is of a high level, similar to other novel drug development programs. Importantly, this was not just a random shot in the dark, instead it was a highly impressive hypothesis driven mission that actually delivered a high-quality compound that is available for further evaluation to prove the hypothesis and potentially deliver a developable asset. It could be argued that in some of the regimen development, drug combination studies in mice, a more factorial approach that builds from 2, to 3 and then perhaps four drugs, with the ability to clearly demonstrate the sterilizing contribution of the asset being evaluated could have been more rigorous than reported here. This leaves this reviewer unable in some cases to confirm that the J bc1 inhibitor contributed anything at all in some of the reported combinations. This could be cleaned up on a re-submission, or in future studies that would attempt to demonstrate the full potential of this asset.

If the work is not original, please provide relevant references. The work is original and very well done

Does the work support the conclusions and claims, or is additional evidence needed? The work supports some of the claims, but as stated above, the usefulness of developing a bc1 inhibitors and that inhibitor demonstrating a real contribution to treatment shortening against a tuberculosis infection still needs more work.

Are there any flaws in the data analysis, interpretation and conclusions? Target validation for bc1 in a highly expressed bd background needs special attention. Contribution of the asset to significant treatment shortening in a developable regimen could be improved

Do these prohibit publication or require revision? I recommend publication of this work with a more substantive discussion of the strain difference and usefulness of bc1 inhibitors vs bd expression and a better demonstration of contribution to treatment shortening

Is the methodology sound? The methods used are sound except for the identified shortcomings

Does the work meet the expected standards in your field? The work is high quality from a highly experienced group working in a target pathway with which they are clearly experts. The regimen development work, especially with this challenging target, relative to bd expression could be improved

Is there enough detail provided in the methods for the work to be reproduced? Yes

Recommendation: Accept for publication with some revision of the highlighted aspects, above

Reviewer #3

(Remarks to the Author)

General Comments

The manuscript presents valuable findings on the role of cytochrome bc1 inhibitors in tuberculosis treatment. However, it currently lacks a clear and well-organized structure, which makes it challenging to follow. To be considered suitable for publication, we strongly recommend rewriting the whole manuscript to include distinct and complete sections—specifically, Introduction, Materials & Methods, Results, Discussion, and Conclusion. The current format hinders readability and clarity and omits critical information necessary to fully assess the study's findings. Additionally, the manuscript would benefit from enhanced experimental justification and a more thorough discussion of the results.

The experimental design, including details on mice strains and compound combinations and its rationale, should be consistently presented across all sections (Introduction, Results, Materials & Methods) and thoroughly discussed.

Introduction

The manuscript provides a well-articulated description of the problem and effectively highlights the potential of the electron transport chain inhibitors in tuberculosis treatment. However, we miss a proper introduction and a clear justification of the two new drug candidates (JNJ-2901 and JNJ-4052) that the authors test (according to the figures), and the differences and their potential benefits when compared to the cytochrome bc1 inhibitor Q203 (Telacebec) already being evaluated in clinical trials. This would provide proof of the manuscript's novelty.

Hypothesis, aim and objectives are not well presented, and clearly insufficiently described. Moreover, supplementary information and figures should not be included in the Introduction but rather integrated and discussed in the results section.

Results

The results demonstrate promising data on JNJ-2901's efficacy in reducing bacterial burden and preventing relapse in mice models. The findings suggest a potential shortening of TB treatment regimens, particularly with BPcJ, which showed the highest efficacy and lowest relapse rates compared to BPcL, BPcM, BPcC, and BPcJ.

However, we have major concerns regarding the description and structure of this section:

- Global experimental plan should first be presented to help understand what the authors did, and then each set of results should be clearly introduced with a summary of the main conclusion before presenting the data of each experimental study.
- The experimental design (mice strain, infection mode, treatment duration, and rationale for each study) and must be clearly stated at the beginning of each results subsection. We here wanted to point out that the naming convention of studies (e.g., Study A, B, C, etc.) does not provide enough information to the reader. We suggest their replacement throughout the manuscript with descriptive terms reflecting the experimental model. For example, instead of "In an additional study (Study C)," we suggest "We additionally investigated drug combination efficacy and relapse in intravenously infected mice."
- Why a second compound – JNJ-4052- is tested in Study F? Why to test another candidate?

Discussion

Although a very small discussion comments are integrated into the manuscript (e.g., L136-143, L156-161, L171-177), a dedicated Discussion section is necessary. We suggest moving the relevant lines into a standalone Discussion section and expanding on several key topics:

- Comparison of JNJ-2901 and JNJ-4052 with Telacebec (Q203): The manuscript should provide a comparative analysis of the new candidates with existing inhibitors in terms of efficacy, safety, and pharmacokinetics as well as treatment efficacy and relapsing rates. What is the novelty? Why these compounds are better? Why they test 2 different compounds? And if everything is done with JNJ-2901, why to include JNJ-4052 in one of the studies?
- Differences in mice models and infection routes: The study utilizes different infection models, and these differences should be considered to discuss the results. In addition, the authors should also discuss the gender bias in the experiments (only female mice are used) and how this could impact clinical translation.
- Comparison between clinical isolates and wild-type (WT) strains: Authors should also discuss the observed differences in drug efficacy between clinical isolates and WT strains. The reasons behind these differences need to be explained and appropriately discussed.
- Experimental limitations: The mention of limitations in L119 is too vague. The authors should explicitly outline the limitations of their experimental design.
- The potential of cytochrome bc1 inhibitors: The claim that these inhibitors could significantly impact TB treatment is overstated for the level of preclinical data presented. Further discussion on the limitations of the preclinical findings hereby presented and next steps for clinical translation is needed, including challenges, barriers and further work to be done before undergoing clinical evaluation.

Materials and Methods

The Materials & Methods section is well-detailed and allows for experimental reproducibility. However, there we have some comments:

- As in the Results section, avoid referring only to different experiments as Study A, B, etc. Instead, add descriptive headings such as "Intranasal Mice Challenge", "Intravenous Mice Challenge", or "High-Dose Aerosol Challenge."
- Each experimental model should consistently include the following information: Mouse strain, sex, and age; inoculum preparation, infection route, and dose; treatment duration and euthanasia time points.
- Authors should provide in the ethical statement the end-point criteria.
- Supplementary tables and results should not be included in the Methods section but should be properly cited in the Results section.
- There are several experimental methods referenced in the supplementary material but not appearing in the main text (e.g PK and tolerability in mouse, ADME assays, HepG2 cytotoxicity assay, mitochondrial toxicity assay Glu/Gal, Ames II Mutagenicity assay, mutant isolation and WGS). These should be briefly mentioned in the main text, such as: "Both JNJ-2901 and JNJ-4052 were evaluated for ADME and toxicity. All parameters supported further use of these compounds in subsequent experiments (Supplementary Table S2)".

Additionally, we suggest to address the following minor comments:

- L62: we would suggest classifying tuberculosis as a global pandemic.
- L79: MoA abbreviation is not needed.

- L87: the word “validated” to describe the inhibitor compound JNJ-2901 is confusing as the presented manuscript seems to be the validation of the compound as an effective treatment adjuvant. If this is not the case, please provide the reference where the compound is validated for its inhibition of the cytochrome bc1.
- L91-96: we suggest to relocate this in the line 76, as the TB-PRACTECAL trial results are a good justification of linezolid-associated adverse effects when treating MDR-TB.
- L116-122: please contextualize the results from this part.
- L127: move the figure citation to line 133. After the results are explained.
- L146: please provide the justification of why using Telacebec in this study and not JNJ-2901 and provide rationale about the drug regimen used for this experimental design.
- L164: as far as we understood, JNJ-2901 and Telacebec were also assessed for its bactericidal activity against clinical isolates in comparison to the H37Rv strain (Figure 2 B,C,D and E). Please provide a proper description of the compound used for this experiment and further discussion of this results in the main text.
- Figure 1: If the mice strain and experimental design are the same, consider re-structure Figure 1B and 1C in a unique figure.
- Figure 2: Please, consider a figure rearrangement. We suggest dividing Figure 2 in two. Section A on one side (resulting in Figure 2) and sections B, C, D, E and F on the other (resulting in Figure 3 A, B, C, D and E). Since the results represented refer to different experimental questions and require a results section for each one.

Final Recommendation

This study presents valuable preclinical data on cytochrome bc1 inhibitors in TB treatment. However, we have significant concerns regarding the manuscript as it currently stands. It lacks a proper structure, clarity, and experimental justification, and the discussion is entirely insufficient. The manuscript needs to be totally rewritten to address these issues, ensuring that it achieves the required readability, scientific rigor, and overall impact to be considered suitable for publication.

Reviewer #4

(Remarks to the Author)

Version 1:

Reviewer comments:

Reviewer #1

(Remarks to the Author)

The revised manuscript of Clara Aguilar Pérez and colleagues has addressed all of my previous concerns and I am happy to see some of the recommendation on the revised discussion section as well as extended new data. The study represents an important progress on inhibition of respiration machinery in Mtb and has potential for the development of future therapeutics.

Reviewer #3

(Remarks to the Author)

We would like to thank the authors for thoughtfully addressing our major suggestions regarding the manuscript. Our feedback primarily focused on improving its structure, clarity, and the discussion surrounding the experimental procedures and results. In particular, the authors have now effectively justified the differences observed when treating laboratory strains with different cytochrome bc1 inhibitors versus clinical strains. With these revisions, the manuscript now clearly highlights the significance of this investigation in advancing our understanding of tuberculosis treatment strategies.

This work underscores the potential of bc1 inhibitors as viable alternatives to fluoroquinolones and linezolid, which are often associated with adverse effects in current regimens for drug-resistant tuberculosis. Moreover, it contributes to the possibility of treatment shortening for drug-sensitive tuberculosis, with added potential against clinical strains. The experiments were performed using different, well-justified preclinical mouse models, which strengthens the translational relevance of the findings and supports further investigation of this promising asset. Based on these improvements and the scientific merit of the study, we believe the manuscript is now better suited for publication.

However, we would like the authors to address the following remaining issue:

Please add a paragraph on limitations to the discussion section. For example, the sample size of mice used in the experiments could be acknowledged as a limitation (even though we recognize that this is already mentioned on page 7). Additionally, regarding the authors' response that “no differences between sexes have been reported in relation to drug efficacy in TB mouse models,” we would like to stress that, even if ethical requirements do not mandate the inclusion of both biological sexes, doing so is strongly advisable. There is robust evidence from both clinical and preclinical studies that biological sex influences TB epidemiology and treatment outcomes (for example, see: Idris, R., *Infection* 2025, <https://doi.org/10.1007/s15010-024-02424-5>; Dutta NK, *Front Immunol*, 2020, doi: 10.3389/fimmu.2020.01465; Tannenbaum,

C., Nature, 2019, <https://doi.org/10.1038/s41586-019-1657-6>). These differences are driven by a combination of hormonal, genetic, and immunological factors, as well as potential differences in treatment adherence and broader social determinants of health. While we acknowledge that using female mice can be more practical in terms of logistics and animal management- since males often display more aggressive behavior that can complicate experiments- there is now ample evidence showing that men and women often have different disease trajectories (not only for TB!) and responses to treatment. One persistent issue in drug development is that preclinical and clinical studies frequently fail to reflect the biological diversity found in real-world populations. Including both sexes in preclinical experiments wherever feasible would help improve the generalizability of the findings and ensure that results more accurately inform treatment strategies for all patients.

In summary, we encourage the authors to carefully consider any additional limitations of their study and clearly state them, as this will add valuable context and transparency. Once this is done, the manuscript can be endorsed for publication.

Reviewer #4

(Remarks to the Author)

Reviewers' comments

We thank the reviewers for providing useful comments and suggestions that have greatly improved this manuscript. We have addressed each comment below (red).

Reviewer #1 (Remarks to the Author):

The manuscript by Perez and colleagues present data on the effects of cytochrome bc₁ inhibitors and their potential efficacy in drug regimens against multidrug resistant TB as well as its role in reducing the duration of chemotherapy in drug susceptible TB. The work is important and timely considered the burden of drug-resistant TB and the prolonged treatment protocols for both drug resistant and susceptible cases of TB. The experimental approach is solid and data presented are robust to support their hypotheses.

One of the main interesting points in the manuscript is the fact that cytochrome bc₁ inhibitors are more potent against clinical isolates than the lab strains. A further investigation into this would have increased the impact of the work as well as our understanding of the mechanisms of action of these inhibitors. Especially as these are claimed to be validated inhibitors, it would also be nice to see more of their characterisation and target engagement in vitro.

A second paper has recently been published which describes the characterisation, validation and target engagement of the JNJ-2901 inhibitor (PMID:40191462). We have provided validation of JNJ-4052, which comes from the same chemical series as JNJ-2901, in **Supplementary Tables S1-3**.

In addition, would this variability on strain efficacy mean that there are potential other targets being inhibited by the same compounds or is it the secondary effects of inhibiting the cytochrome bc₁ and the respiratory chain of Mtb?

We do not believe that there are any genuine secondary targets for these compounds. Our resistance data demonstrates that a single nucleotide polymorphism in *qcrB* can provide high level resistance (for example T313A leads to >2000-fold resistance to JNJ-4052; **Supplementary Table S1**). As discussed below, we believe the differences are most likely due to the increased expression of cytochrome *bd*, an alternative oxidase that can compensate for cytochrome bc₁ inhibition. This is now outlined in the Discussion section (pg 9-10).

For the present study, authors can expand the discussion regarding the efficacy of their compound in the background of *bd* and how this could affect the outcome. More importantly, the authors can use these compounds to further interrogate this hypothesis. In addition, having another look at their data, I could see that if you calculate the MBC/MIC ratio for the reference lab strain (H37Rv) and the cytochrome deleted mutant (H37Rv_CytBd-KO) on supplementary table S2 one can see that their inhibitor is bacteriostatic on H37Rv whereas it is bactericidal on cytochrome *bd* deleted mutant (H37Rv_CytBd-KO). That further adds to the argument that the expression levels of cytochrome *bd* is an important factor in their compound efficacy.

We agree that this is an important point and have now provided discussion of the difference in bactericidal activity between H37Rv and H37Rv_CytBd-KO strains in the Discussion section (pg 9-10).

Furthermore, the authors did not include if these compounds are active against MDR and XDR Mtb strains. Taking into account that the main use of these inhibitors would be towards a drug regimen for treating MDR-TB, it would be nice to see that the compounds maintain their activity against a panel of multiple drug-resistant strains including MDR and XDR clinical isolates.

Activity of JNJ-2901 against a panel of 18 MDR-TB strains has now been published showing that the inhibitor is more active compared to lab-adapted strains (PMID:40191462).

Obviously a series of experiments where one looks at the levels of expression of these genes under treatment or not in a series of lab and clinical strains of Mtb would shed light on this question. But I don't think these experiments need to be part of this work.

Lastly, in the Figure S3, I cannot see the control line as well as the figure legend need a bit more explanation.

We have updated the figure legend to indicate that moxifloxacin is hidden by BDQ (**Supplementary Figure S3**). We have added more detail of the experimental conditions in the figure legend.

Reviewer #2 (Remarks to the Author):

The authors have done an excellent job of identifying cytochrome bc1 as an important component of the Mtb electron transport chain and a potentially sterilizing target in for TB regimen development. Using a relapsing mouse model, they have attempted to demonstrate that cytochrome bc1 inhibitors could be effective partner drugs in TB regimen development that could enhance sterilization. They report on several novel regimens as examples of this role for bc1 inhibitors in regimens for both multidrug-resistant TB (MDR-TB) and drug-sensitive TB (DS-TB), where cytochrome bc1 inhibitors could contribute to sterilization and treatment shortening. importantly, they have shown that clinical isolates exhibit heightened susceptibility to cytochrome bc1 inhibitors compared to laboratory-adapted strains, further supporting their potential usefulness in a clinical setting even as their contribution may be low or undetectable against a lab adapted strain. They take these findings as confirmation that cytochrome bc1 inhibitors have significant potential to improve TB treatment outcomes and highlight the need for further studies to evaluate their clinical contribution to novel treatment regimens.

What are the noteworthy results? The authors have done an excellent job of identifying and advancing highly potent inhibitors against their chosen target, cytochrome bc1. There is enough evidence to demonstrate that this is a sound choice of target, especially with the excellent success of bedaquiline, the flagship ATP synthase developed by the same group that has literally changed the paradigm in TB treatment shortening and highlighted the electron transport chain as a highly valuable and sterilizing target, at the same level or better than the rifamycins of prior years. It is therefore sound and admirable to hit this pathway as hard as possible, at multiple points and potentially achieve better cure. The problem seems to be that the interplay between cytochrome bc and cytochrome bd seems complicated and makes it difficult to determine with available data, including the data reported here, if just inhibiting bc, but not bd will be enough to realize the expected benefits. It seems that high expression of bd is able to reverse or nullify the effects of inhibiting bc. This is potentially the reason why there is strain difference between lab adapted versus clinical isolates. Further proof is gleaned from the excellent performance of Q203, the lead asset in this target space, versus mycobacterial species that do not express bd like *M. leprae* and *M. ulcerans*. This leaves the potential for a bc inhibitor uncertain against tuberculosis. Having said that, the team has identified a tool compound that together with Q203 will be useful in further investigating this target

Will the work be of significance to the field and related fields? While the actual developability of this asset in future TB drug regimens is in doubt until more work is done to fully appreciate contribution

against all TB strains, publication of this work will be of great significance to the field and specifically to the grand idea of targeting the electron transport chain of Mtb at multiple nodes to achieve better cure for TB. Furthermore, the compound reported here could also be expanded to the NTM space as is being done for Q203 which is currently being evaluated against leprosy and buruli ulcer, arguably important and neglected diseases as well. But for use in tuberculosis, more work needs to be done to understand the importance of targeting bc1 in a background with high expression of bd. And to determine if the site of infection and location in a lesion affects the usefulness of this asset. And if the mouse model as used today is the best model for this particular asset

How does it compare to the established literature? The work reported here is of a high level, similar to other novel drug development programs. Importantly, this was not just a random shot in the dark, instead it was a highly impressive hypothesis driven mission that actually delivered a high-quality compound that is available for further evaluation to prove the hypothesis and potentially deliver a developable asset. It could be argued that in some of the regimen development, drug combination studies in mice, a more factorial approach that builds from 2, to 3 and then perhaps four drugs, with the ability to clearly demonstrate the sterilizing contribution of the asset being evaluated could have been more rigorous than reported here. This leaves this reviewer unable in some cases to confirm that the J bc1 inhibitor contributed anything at all in some of the reported combinations. This could be cleaned up on a re-submission, or in future studies that would attempt to demonstrate the full potential of this asset.

A more stepwise factorial approach (building from two- to three- and then four-drug combinations) could have further clarified the sterilising contribution of a cytochrome *bc₁* inhibitor. However, the aim of these studies was not to develop an entirely new regimen but rather to replace moxifloxacin and linezolid with a potentially superior compound while maintaining the existing backbone. Therefore, we prioritised evaluating the *bc₁* inhibitor in combination with a clinically relevant regimen to reflect a realistic therapeutic setting.

Additionally, we were mindful of the ethical considerations regarding animal use. Expanding the study design to include a larger number of combinations/conditions would have significantly increased the number of animals required.

In several of our studies, the addition of a cytochrome *bc₁* inhibitor led to a statistical improvement in relapse rates (e.g. **Figure 2**; CZ vs CZT). Nevertheless, we agree that future investigations could employ a more granular factorial design to better define the specific contribution of the *bc₁* inhibitor. These follow-up studies would help further demonstrate the full potential of cytochrome *bc₁* inhibitors and strengthen its case for clinical development.

If the work is not original, please provide relevant references. The work is original and very well done

Does the work support the conclusions and claims, or is additional evidence needed? The work supports some of the claims, but as stated above, the usefulness of developing a bc1 inhibitors and that inhibitor demonstrating a real contribution to treatment shortening against a tuberculosis infection still needs more work.

While we acknowledge that additional studies are needed to fully establish the potential of cytochrome *bc1* inhibitors to shorten treatment duration, particularly in the context of multidrug-resistant tuberculosis (MDR-TB), the data presented here clearly demonstrate that the inclusion of a cytochrome *bc₁* inhibitor reduces the treatment time required. For example, the addition of telacebec to CZ reduces the treatment by at least 2 months (**Figure 2**). This is the starting point for further work to investigate this in more detail.

Are there any flaws in the data analysis, interpretation and conclusions? Target validation for bc1 in a highly expressed bd background needs special attention.

We have recently published the validation of JNJ-2901 (PMID: 40191462) and have expanded our discussion on bd expression (pg 9-10).

Contribution of the asset to significant treatment shortening in a developable regimen could be improved

As discussed above, we view this work as an important starting point for investigating the potential contribution of cytochrome *bc*₁ inhibitors to future treatment-shortening regimens. We agree that further work is required beyond the current study. However, one challenge in addressing this more definitively is the lack of consensus on what constitutes a 'developable' regimen, as different organisations often have varying perspectives on which drugs should be prioritised; clofazimine being a notable example here.

Do these prohibit publication or require revision? I recommend publication of this work with a more substantive discussion of the strain difference and usefulness of bc1 inhibitors vs bd expression and a better demonstration of contribution to treatment shortening

We have now re-formatted the manuscript and expanded the discussion on the clinical isolate differences we observed.

Is the methodology sound? The methods used are sound except for the identified shortcomings

Does the work meet the expected standards in your field? The work is high quality from a highly experienced group working in a target pathway with which they are clearly experts. The regimen development work, especially with this challenging target, relative to bd expression could be improved

Is there enough detail provided in the methods for the work to be reproduced? Yes

Recommendation: Accept for publication with some revision of the highlighted aspects, above

Reviewer #3 and #4 (Remarks to the Author):

General Comments

The manuscript presents valuable findings on the role of cytochrome bc1 inhibitors in tuberculosis treatment. However, it currently lacks a clear and well-organized structure, which makes it challenging to follow. To be considered suitable for publication, we strongly recommend rewriting the whole manuscript to include distinct and complete sections—specifically, Introduction, Materials & Methods, Results, Discussion, and Conclusion. The current format hinders readability and clarity and omits critical information necessary to fully assess the study's findings. Additionally, the manuscript would benefit from enhanced experimental justification and a more thorough discussion of the results.

We have now re-formatted the manuscript for Nature Communications as suggested and expanded on the Discussion section.

The experimental design, including details on mice strains and compound combinations and its

rationale, should be consistently presented across all sections (Introduction, Results, Materials & Methods) and thoroughly discussed.

Introduction

The manuscript provides a well-articulated description of the problem and effectively highlights the potential of the electron transport chain inhibitors in tuberculosis treatment. However, we miss a proper introduction and a clear justification of the two new drug candidates (JNJ-2901 and JNJ-4052) that the authors test (according to the figures), and the differences and their potential benefits when compared to the cytochrome bc₁ inhibitor Q203 (Telacebec) already being evaluated in clinical trials. This would provide proof of the manuscript's novelty.

The primary focus of this manuscript is to understand the contribution of a cytochrome *bc*₁-targeting inhibitor to TB treatment regimens, rather than specifically focusing on JNJ-2901 or suggesting it is a superior molecule. We consider all the compounds discussed in this work as tool compounds used to increase our understanding of how this mode of action could complement existing drug regimens. In parallel, we have published a separate validation of JNJ-2901 (PMID:40191462) and have added additional justification here (pg 5-6).

Hypothesis, aim and objectives are not well presented, and clearly insufficiently described.

We have clarified and expanded the objectives of each section as requested.

Moreover, supplementary information and figures should not be included in the Introduction but rather integrated and discussed in the results section.

Reference to the figures and supplementary information has been removed from the introduction following re-formatting.

Results

The results demonstrate promising data on JNJ-2901's efficacy in reducing bacterial burden and preventing relapse in mice models. The findings suggest a potential shortening of TB treatment regimens, particularly with BPaCJ, which showed the highest efficacy and lowest relapse rates compared to BPaL, BPaM, BPaC, and BPaJ.

However, we have major concerns regarding the description and structure of this section:

- Global experimental plan should first be presented to help understand what the authors did, and then each set of results should be clearly introduced with a summary of the main conclusion before presenting the data of each experimental study.

We agree, and have now divided this section more clearly with **Figures 1b-c** now discussed sequentially. An overview of the first set of experiments is provided in **Figure 1a**.

- The experimental design (mice strain, infection mode, treatment duration, and rationale for each study) and must be clearly stated at the beginning of each results subsection. We here wanted to point out that the naming convention of studies (e.g., Study A, B, C, etc.) does not provide enough information to the reader. We suggest their replacement throughout the manuscript with descriptive terms reflecting the experimental model. For example, instead of "In an additional study (Study C)," we suggest "We additionally investigated drug combination efficacy and relapse in intravenously infected mice."

We respectfully disagree, as the experimental design is indicated in the Materials and Methods section. We do not believe it is necessary to repeat this information in the Results section or figure legends. We have named them A, B, C etc. because the studies were performed by different institutions rather than because they are different type of studies.

-Why a second compound – JNJ-4052- is tested in Study F? Why to test another candidate?

JNJ-2901 and JNJ-4052 belong to the same chemical series and exhibit very similar activity and safety profiles. Our cytochrome *bc*₁ drug discovery programme has been ongoing for 7 years, during which time we have progressed a number of lead compounds including JNJ-4052 and JNJ-2901. The *in vivo* clinical isolate study (Study F) was performed using the earlier lead, JNJ-4052. While we acknowledge that repeating this experiment with JNJ-2901 would have been ideal, given the comparable profiles of both compounds, we do not anticipate a change in the outcome. Therefore, we believe repeating this study would not represent a justifiable use of mice.

Discussion

Although a very small discussion comments are integrated into the manuscript (e.g., L136-143, L156-161, L171-177), a dedicated Discussion section is necessary. We suggest moving the relevant lines into a standalone Discussion section and expanding on several key topics:

- Comparison of JNJ-2901 and JNJ-4052 with Telacebec (Q203): The manuscript should provide a comparative analysis of the new candidates with existing inhibitors in terms of efficacy, safety, and pharmacokinetics as well as treatment efficacy and relapsing rates. What is the novelty? Why these compounds are better? Why they test 2 different compounds? And if everything is done with JNJ-2901, why to include JNJ-4052 in one of the studies?

The aim of the project was to investigate the role of cytochrome *bc*₁ inhibitors in future treatment regimens rather than suggest that JNJ-2901 is superior to telacebec. We selected three inhibitors with related chemical structure, mode of binding and efficacy to achieve these aims; we have made this clearer in the text (pg 5-6).

- Differences in mice models and infection routes: The study utilizes different infection models, and these differences should be considered to discuss the results. In addition, the authors should also discuss the gender bias in the experiments (only female mice are used) and how this could impact clinical translation.

In this study, two different infection models were used: nasal (via aerosol or intranasal instillation) and intravenous. As noted in the text, we acknowledge that the intravenous model exhibits a higher severity of the disease since this infection route confers a systemic infection, whereas the nasal route mainly leads to a pulmonary infection. The intention of this study was not to characterise the treatment response at the pathology level, but rather to have a broad knowledge of the activity of cytochrome *bc*₁-containing regimens in multiple potential scenarios.

Regarding the use of only female mice, we acknowledge recommendations for including both sexes in animal studies. However, in line with our institutional policies focused on the 3Rs, we used only female mice to minimise stress and harm. Relapse studies are very long and involve repeated daily handling and dosing, which can lead to a higher risk of aggress behaviour in males. Clinical translation would take place when human exposures are known in a Phase I trial. Additionally, no differences between sex have been reported in relation to drug efficacy in TB mouse models.

- Comparison between clinical isolates and wild-type (WT) strains: Authors should also discuss the

observed differences in drug efficacy between clinical isolates and WT strains. The reasons behind these differences need to be explained and appropriately discussed.

We have now added discussion on the comparison of WT vs. clinical isolate (pg 9-10).

- Experimental limitations: The mention of limitations in L119 is too vague. The authors should explicitly outline the limitations of their experimental design.

The main limitation of this study was the number of animals used did not provide enough statistical power to differentiate the different conditions. We have mentioned this in the text (pg 7).

- The potential of cytochrome bc1 inhibitors: The claim that these inhibitors could significantly impact TB treatment is overstated for the level of preclinical data presented. Further discussion on the limitations of the preclinical findings hereby presented and next steps for clinical translation is needed, including challenges, barriers and further work to be done before undergoing clinical evaluation.

As discussed above, we wish to make it clear that JNJ-2901 is a tool compound that we have used to demonstrate how, inhibitors of cytochrome *bc*₁, a novel mode of action, would interact with existing treatments. Our conclusions refer to 'cytochrome *bc*₁ inhibitors' rather than to a specific compound in development.

Materials and Methods

The Materials & Methods section is well-detailed and allows for experimental reproducibility.

However, there we have some comments:

- As in the Results section, avoid referring only to different experiments as Study A, B, etc. Instead, add descriptive headings such as "Intranasal Mice Challenge", "Intravenous Mice Challenge", or "High-Dose Aerosol Challenge."

- Each experimental model should consistently include the following information: Mouse strain, sex, and age; inoculum preparation, infection route, and dose; treatment duration and euthanasia time points.

We respectfully disagree with this suggestion. The experimental details for each study are already included in the Methods section, and adding them to the Results and figure legends would lead to unnecessary repetition. We have intentionally used this naming strategy to clearly distinguish studies from different institutions. Furthermore, many of the models have similar experimental parameters, making it difficult to differentiate them by name alone.

- Authors should provide in the ethical statement the end-point criteria.

We have now included the humane end points for the *in vivo* studies (pg 18).

- Supplementary tables and results should not be included in the Methods section but should be properly cited in the Results section.

Where possible, we have discussed supplementary tables and results in the Results section, however we disagree that these elements cannot be cited in the methods.

- There are several experimental methods referenced in the supplementary material but not appearing in the main text (e.g PK and tolerability in mouse, ADME assays, HepG2 cytotoxicity assay, mitochondrial toxicity assay Glu/Gal, Ames II Mutagenicity assay, mutant isolation and WGS). These should be briefly mentioned in the main text, such as: "Both JNJ-2901 and JNJ-4052

were evaluated for ADME and toxicity. All parameters supported further use of these compounds in subsequent experiments (Supplementary Table S2)".

We have now included this information in the Results section (pg 5-6).

Additionally, we suggest to address the following minor comments:

- L62: we would suggest classifying tuberculosis as a global pandemic.

We have updated this as suggested.

- L79: MoA abbreviation is not needed.

'MoA' is used elsewhere in the manuscript.

- L87: the word "validated" to describe the inhibitor compound JNJ-2901 is confusing as the presented manuscript seems to be the validation of the compound as an effective treatment adjuvant. If this is not the case, please provide the reference where the compound is validated for its inhibition of the cytochrome bc1.

We have kept 'validated' as we can now present an additional manuscript (PMID:40191462), which outlines this validation.

- L91-96: we suggest to relocate this in the line 76, as the TB-PRACTECAL trial results are a good justification of linezolid-associated adverse effects when treating MDR-TB.

We have now moved this section to the Introduction as suggested.

- L116-122: please contextualize the results from this part.

We have expanded the results as requested.

- L127: move the figure citation to line 133. After the results are explained.

This has been moved as requested.

- L146: please provide the justification of why using Telacebec in this study and not JNJ-2901 and provide rationale about the drug regimen used for this experimental design.

This study was inspired by previous *in vitro* work demonstrating that the combination of bedaquiline, clofazimine, and telacebec resulted in rapid killing (PMID:27506290). We aimed to evaluate the *in vivo* efficacy of treatment regimens containing these compounds in combination with pyrazinamide, which has also been shown to enhance efficacy, potentially supporting treatment shortening (PMID:25622149). To allow comparison with previous work, we continued to use telacebec in this study rather than JNJ-2901. We have now clarified this justification in the results (pg 7) and discussion (pg 9).

- L164: as far as we understood, JNJ-2901 and Telacebec were also assessed for its bactericidal activity against clinical isolates in comparison to the H37Rv strain (Figure 2 B,C,D and E). Please provide a proper description of the compound used for this experiment and further discussion of this results in the main text.

At the time that this experiment was initiated, JNJ-4052 was our lead molecule from our cytochrome bc1 drug discovery programme. This was the main reason it was chosen for the *in vivo* study outlined in **Figure 3e**. This compound provides a proof-of-principle for inhibition of cytochrome bc1, with other compounds with this MoA expected to have the same response as suggested by **Figure 3b-d**.

- Figure 1: If the mice strain and experimental design are the same, consider re-structure Figure 1B and 1C in a unique figure.

We would prefer to keep these studies separate, enabling the reader to discriminate between the individual experiments.

- Figure 2: Please, consider a figure rearrangement. We suggest dividing Figure 2 in two. Section A on one side (resulting in Figure 2) and sections B, C, D, E and F on the other (resulting in Figure 3 A, B, C, D and E). Since the results represented refer to different experimental questions and require a results section for each one.

We agree, and have now split **Figure 2** into two new figures as suggested.

Final Recommendation

This study presents valuable preclinical data on cytochrome bc1 inhibitors in TB treatment. However, we have significant concerns regarding the manuscript as it currently stands. It lacks a proper structure, clarity, and experimental justification, and the discussion is entirely insufficient. The manuscript needs to be totally rewritten to address these issues, ensuring that it achieves the required readability, scientific rigor, and overall impact to be considered suitable for publication.

Response to reviewer's comments

We have addressed the final reviewer's comment in red below. We once again thank the reviewer's for their valuable comments.

Reviewer #1 (Remarks to the Author):

The revised manuscript of Clara Aguilar Pérez and colleagues has addressed all of my previous concerns and I am happy to see some of the recommendation on the revised discussion section as well as extended new data. The study represents an important progress on inhibition of respiration machinery in Mtb and has potential for the development of future therapeutics.

Reviewer #3 (Remarks to the Author):

We would like to thank the authors for thoughtfully addressing our major suggestions regarding the manuscript. Our feedback primarily focused on improving its structure, clarity, and the discussion surrounding the experimental procedures and results. In particular, the authors have now effectively justified the differences observed when treating laboratory strains with different cytochrome bc1 inhibitors versus clinical strains. With these revisions, the manuscript now clearly highlights the significance of this investigation in advancing our understanding of tuberculosis treatment strategies.

This work underscores the potential of bc1 inhibitors as viable alternatives to fluoroquinolones and linezolid, which are often associated with adverse effects in current regimens for drug-resistant tuberculosis. Moreover, it contributes to the possibility of treatment shortening for drug-sensitive tuberculosis, with added potential against clinical strains. The experiments were performed using different, well-justified preclinical mouse models, which strengthens the translational relevance of the findings and supports further investigation of this promising asset. Based on these improvements and the scientific merit of the study, we believe the manuscript is now better suited for publication.

However, we would like the authors to address the following remaining issue:

Please add a paragraph on limitations to the discussion section. For example, the sample size of mice used in the experiments could be acknowledged as a limitation (even though we recognize that this is already mentioned on page 7). Additionally, regarding the authors' response that "no differences between sexes have been reported in relation to drug efficacy in TB mouse models," we would like to stress that, even if ethical requirements do not mandate the inclusion of both biological sexes, doing so is strongly advisable. There is robust evidence from both clinical and preclinical studies that biological sex influences TB epidemiology and treatment outcomes (for example, see: Idris, R., *Infection* 2025, <https://doi.org/10.1007/s15010-024-02424-5>; Dutta NK, *Front Immunol*, 2020, doi: 10.3389/fimmu.2020.01465; Tannenbaum, C., *Nature*, 2019, <https://doi.org/10.1038/s41586-019-1657-6>). These differences are driven by a combination of hormonal, genetic, and immunological factors, as well as potential differences in treatment adherence and broader social determinants of health. While we acknowledge that using female mice can be more

practical in terms of logistics and animal management- since males often display more aggressive behavior that can complicate experiments- there is now ample evidence showing that men and women often have different disease trajectories (not only for TB!) and responses to treatment. One persistent issue in drug development is that preclinical and clinical studies frequently fail to reflect the biological diversity found in real-world populations. Including both sexes in preclinical experiments wherever feasible would help improve the generalizability of the findings and ensure that results more accurately inform treatment strategies for all patients.

We thank the reviewer for their helpful suggestion. We have now added a paragraph to the discussion (pg. 10) outlining the limitations of our study and lessons for future work. Specifically, we address the use of female mice, as suggested, and also note the following points for improvement of future studies: (i) increasing statistical power by using larger animal group sizes, (ii) incorporating monotherapy arms to enable regimen deconvolution, and (iii) including clinical isolates to better reflect the diversity and relevance of circulating *M. tuberculosis* strains.

In summary, we encourage the authors to carefully consider any additional limitations of their study and clearly state them, as this will add valuable context and transparency. Once this is done, the manuscript can be endorsed for publication.

Reviewer #4 (Remarks to the Author):
